# Recovery of Bioactive Compounds from Marine Organisms: Focus on the Future Perspectives for Pharmacological, Biomedical and Regenerative Medicine Applications of Marine Collagen

**DOI:** 10.3390/molecules28031152

**Published:** 2023-01-24

**Authors:** Salvatrice Rigogliuso, Simona Campora, Monica Notarbartolo, Giulio Ghersi

**Affiliations:** 1Department of Biological, Chemical and Pharmaceutical Sciences and Technologies (STEBICEF), University of Palermo, Viale delle Scienze, Ed. 16, 90128 Palermo, Italy; 2Abiel s.r.l., c/o Department STEBICEF, University of Palermo, Viale delle Scienze, Ed. 16, 90128 Palermo, Italy

**Keywords:** marine collagen, jellyfish, chondrocyte maintenance, cell therapy, tissue regeneration

## Abstract

Marine environments cover more than 70% of the Earth’s surface and are among the richest and most complex ecosystems. In terms of biodiversity, the ocean represents an important source, still not widely exploited, of bioactive products derived from species of bacteria, plants, and animals. However, global warming, in combination with multiple anthropogenic practices, represents a serious environmental problem that has led to an increase in gelatinous zooplankton, a phenomenon referred to as jellyfish bloom. In recent years, the idea of “sustainable development” has emerged as one of the essential elements of green-economy initiatives; therefore, the marine environment has been re-evaluated and considered an important biological resource. Several bioactive compounds of marine origin are being studied, and among these, marine collagen represents one of the most attractive bio-resources, given its use in various disciplines, such as clinical applications, cosmetics, the food sector, and many other industrial applications. This review aims to provide a current overview of marine collagen applications in the pharmacological and biomedical fields, regenerative medicine, and cell therapy.

## 1. Introduction

The oceans are likely the Earth’s most valuable natural food resource, providing mainly fish and shellfish. However, global climate change, in combination with multiple anthropogenic practices, such as the development of the coasts, the increase in marine culture operations with the consequent increase in nutrients, pollution, eutrophication, and ocean acidification, represents a serious environmental problem [1,2]. The Mediterranean sea, with its long history of overfishing, is the home of numerous species of marine organisms, including jellyfish, both native and invasive. Overfishing is an important cause of increases in jellyfish populations, due to the removal of predators and competitors of jellyfish [3]. Moreover, the disposal of large amounts of fishing-industry waste presents several environmental waste management problems. All these factors have a significant ecological and socio-economic impact, and typically negatively affect human health and activities in coastal waters [4]. Fortunately, what was previously only considered to be a serious environmental issue, has been largely re-evaluated in recent years, and the marine environment is now considered as a new alternative source from which to recover important biological compounds [4,5]. Because of its phenomenal biodiversity, the marine world is a rich natural resource for many biologically active compounds. Marine organisms live in complex habitats and are exposed to extreme conditions, and thus produce a wide variety of specific and potent active substances that cannot be found elsewhere [6]. Many marine compounds have been detected as having various biological activities: peptides isolated from fish and algal polysaccharides have been reported to have anticancer, anticoagulant, and antihypercholesterolemic activities [7]. Marine bacteria and fish oils contain high levels of omega-3 fatty acids, whereas seaweed and shellfish such as crustaceans contain potent antioxidants, including carotenoids and phenolic compounds [8]. The marine by-products themselves also represent an interesting source of collagen, a fibrous protein with great social and economic potential for a wide range of biomedical, pharmaceutical, and cosmetic applications [9]. In recent years, environmental and economic concerns have encouraged the use of eco-friendly alternatives the exploitation of the Earth’s natural resources. Therefore, the European Commission has approved “Blue Growth” as a long-term strategy to support sustainable development in the marine sectors [10]. In the last fifteen years, the literature on the possible applications of marine collagen has grown enormously (Figure 1). This reflects the growing concern of scientists over protection of the marine environment and the willingness to exploit its enormous resources.

## 2. Food and Human Health

It is well known that regular consumption of fish and seafood is associated with beneficial health effects. The close relationship between nutrition and health was already known in the mid-1980s, and this has led to the development of the concept of functional foods. Functional foods, such as pre- and pro-biotics, and cholesterol-lowering products, can improve the general condition of organisms, and can decrease the risk of, or even cure, some diseases, such as cardiovascular disease and osteoporosis [6,11]. The Mediterranean food model is a highly regarded nutritional model and seafood is a very important constituent of the Mediterranean diet. Epidemiological studies have shown that people in the Mediterranean Sea region live longer, and have lower chronic disease rates, than inland dwellers [12,13]. In recent years, consumer interest in the nutritional aspects of food has grown enormously. The ability to identify healthy components in food and to promote this information enables consumers to make healthier food choices. The concept of functional foods, able to prevent diet-related diseases and improve the physical and mental welfare of consumers, has therefore become popular [14]. Based on this concept, marine products are considered excellent functional foods as they are rich in protein, containing all essential amino acids, have well-known antioxidant properties, high levels of polyunsaturated fatty acids (omega-3 and omega-6), minerals such as Ca and I, vitamins, especially those in group B (B1, B2 and B12), and many other nutrients [5,6]. Polyunsaturated fatty acids, particularly eicosapentaenoic acid and docosahexaenoic acid, are considered to be “good fats”, play important roles in the modulation of many physiological processes, and are more significant than other sources of animal protein [15]. Seafood, such as various species of fish, mollusks, and crustaceans, contains significant amounts of these “good” fats, and also provides high-quality protein, including all of the essential dietary amino acids required for the maintenance and growth of the human body [16]. Epidemiological studies and clinical trials have shown a relationship between the intake of omega-3 and its beneficial effects on different diseases; it plays a key role in the prevention and treatment of many chronic diseases, such as neurological and cardiovascular disorders [17], cancer (breast, colorectal, prostate, etc.), inflammatory diseases, asthma, inflammatory bowel disease, rheumatoid arthritis [18], psoriasis, osteoporosis [19], obesity, and diabetes mellitus [20] (Figure 2). Furthermore, much attention has been recently paid to the many natural bioactive compounds that can be obtained from marine organisms. 

## 3. Sources of Marine Bioactive Compounds

Most of the natural products extracted from marine organisms have several important applications for humans [21]; many of these are derived from benthic cnidarians, such as sea anemones and corals, while a limited number of bioactive compounds have been extracted from pelagic cnidarians (hydromedusae and scyphomedusae). Corals, in particular, have a long evolutionary history of developing proteins and genes that govern biomineralization, many of which are highly conserved and analogous to human variants [22]. Despite being a potential source of bioactive molecules, the use of coral extracts is limited and mainly focused on their potential use as scaffolds for grafting procedures, given their qualitative and structural similarity to human bone [21,22]. Marine sponges are another ancestral group of animals that have been extensively studied regarding their biological role in marine ecosystems, and furthermore, as an untapped source of bioactive compounds, in particular, spongin-like collagen [23]. Collagen is probably one of the most important components obtained as a result of the re-evaluation of marine sources; it can be isolated from marine invertebrates, such as sea urchins, squid, prawns, starfishes [24,25] and marine vertebrates such as fish and marine mammals, or other marine sources such as algae [26] (Figure 3). Marine collagen extracted from jellyfish may become the main biological component of objects such as paper. Jelly-derived paper may be included in the recycling of wood-derived paper [27]. “Green” protocols can be used to extract collagen and Q-mucin glycoproteins from jellyfish; these compounds are suitable for the production of highly biodegradable plastics whose properties can be enhanced using cross-linkers derived from renewable resources. Given that pollution due to plastics is an important issue for current societies, the environmentally friendly, sustainable production of highly biodegradable plastics may have a large variety of applications in the present and the future [28]. In addition to biotechnological applications that aim to reduce environmental impacts, marine resources can find ample space in the biomedical field. One group of marine invertebrates that shows promise as a source of bioactive compounds is algae, particularly macroalgae. Aquamin is a food supplement derived from the red algae *Lithothamnion corallioides*, and contains calcium, magnesium and 72 other trace minerals [29]. *L. Corallioides* is unique, in that it is one of the few algal species to produce a calcareous skeleton. Aquamin has been tested, and its effects appear to be related to the increased mineralization of a pre-osteoblastic cell line in vitro, to bone mineralization, with an increased mineral/matrix ratio, and to the hydroxyapatite content of the trabecular bone in vivo [30]. Within the algal group, fucoidan is one of the best-studied extracts. Fucoidans are highly sulphated and fucose-rich polymers, found in a heavily branched form in brown macroalgae, and in a more linear form in echinoderms. These marine polymers are multifunctional, with a range of therapeutic uses, from anti-inflammatory to antiviral purposes [31]. Among many natural polymers, including chitin and alginates, collagen is one of the most extensively studied and applied polymers in clinical settings, because of its properties [32]. It plays a fundamental role in the formation of tissues and organs and it is involved in various functional and structural aspects of cells. Excellent biocompatibility and safety for many applications, such as wound healing, cartilage regeneration, skin care, and others, due to its biological characteristics, such as biodegradability and weak antigenicity, make collagen the primary resource for biotechnological applications [33], both in tissue engineering in the biomedical fields [34], and in the cosmetic, nutraceutical and pharmacological industries [35].

## 4. Collagen Characterization

Collagen is the most abundant mammalian protein, accounting for approximately 30% of total body protein. It is the main fibrous structural protein of the extracellular matrix (ECM) and connective tissue of vertebrates, in which it plays a predominantly mechanical function [36,37]. The general structure is characterized by the repetition of “Glycine-X-Y” domains, where “X” and “Y” are occupied by different imino acids that vary throughout the triple helix, though “X” is frequently proline (Pro) and “Y” is frequently hydroxyproline (Hyp) [38]. This pattern leads to the formation of three parallel polypeptide α-chains, forming a triple-helix structure, and giving rise to 28 types of collagens [39,40]. In addition to being the main structural element of the extracellular matrix of human tissues, type I collagen also has numerous intrinsic properties, such as biocompatibility for many biological applications, bioactivity, biodegradability, and the ability, in its native form, to generate hydrogels, by a simple pH change [41,42,43,44,45,46]. Due to these important characteristics, it receives great attention in the biomedical field, both as a functional ingredient in different formulations and as a component of several products, such as medical devices and pharmaceutical systems [46,47,48,49]. Although collagen appeared in the first phase of evolution in primitive animals such as jellyfish, corals and sea anemones, the mammalian (especially bovine and porcine) form is the main source used in the industrial and biomedical fields [50,51,52]. Although both bovine and porcine sources present multiple advantages, there are some disadvantages concerning adverse inflammatory and immunologic responses. For example, almost 3% of the population is allergic to bovine collagen, limiting its use in biomedical applications [53]. On the other hand, even though porcine collagen does not cause intensive allergic responses thanks to its similarity to the human form, many problems and obstacles are due to the possibility of transmitting some zoonoses to humans, bovine spongiform encephalopathy, foot-and-mouth disease [53,54,55,56], as well as restrictions based on religion: bovine collagen is prohibited in Hinduism and porcine collagen is prohibited in Islamic and Jewish cultures, due to religious beliefs [57]. A possible alternative could be the production of collagen in a recombinant form, despite the difficulties resulting from the complex post-translational modifications to which collagen is subject, which makes this system very expensive [58,59]. Several biotech companies are developing innovative platforms for the production of recombinant collagen without using animal sources; among these CollPlant is a biotech company which has succeeded in developing a plant platform that effectively expresses human collagen [60]. The production of recombinant human collagen type I (rhCollagen) was obtained from genetically modified tobacco plants. Recombinant human protein, in form of “procollagen”, was extracted from mature plant leaves and further processed into highly purified rhCollagen, which can be used for manufacturing medical products. Currently, there are two commercial products on the European market based on its rhCollagen technology, developed by CollPlant: an advanced wound care device, indicated for the management of acute and chronic wounds, and a soft-tissue repair matrix, indicated for the treatment of tendinopathies. Evonik is another biotech company that has developed an advanced recombinant collagen platform, called VECOLLAN^®^, made through precisely controlled conditions via an established fermentation-based process. In addition to being sustainable and produced without any use of animals, this process offers a high level of purity that is easily reproducible for use in the biomedical, pharmaceutical, and cosmetic fields. Many tests have shown that recombinant collagens behave like their native counterparts; however, recombinant collagen technology still has many limitations and is currently underutilized by the clinical, commercial and food industries [61]. Marine species can be used as an alternative and safe source of new bioactive peptides. As mentioned above, marine collagen extraction can be obtained from both invertebrates and vertebrates, such as sponges, coralline red algae, sea urchins, octopi, squid, jellyfishes, cuttlefishes, starfishes, sea anemones, and prawns, in addition to different species of fishes [62]. Among marine invertebrates, jellyfish contain approximately 95–99% water by weight [63], while minerals (15.9–57.2 g/100 g) and protein dry weight (20.0–53.9 g/100 g) were found to be the richest components; specifically, Khong and coworkers found that the total collagen content of jellyfish varies from 122.64 to 693.92 mg/g dry weight, accounting for approximately half the total protein content [64], and that collagen derived from *Rhizostoma pulmo* has been shown to have a high degree of similarity to mammalian type I collagen [65]. Jellyfish collagen shares several features with its vertebrate counterparts [24]: it shows high potential in bone and cartilage regeneration, where it is highly biocompatible and bioinductive when used as a scaffold [66,67], and in wound healing to attract fibroblasts and promotes the deposition of new collagen in the wound bed [68,69]. Jeyllyfish collagen also has immunostimulatory effects on cultured cells [70], biological activities such as angiotensin-converting enzyme inhibitory action [71], and antifatigue [72] and antioxidant proprieties [71], and therefore, it may be used as a biomaterial for prostheses, encapsulating polymers for drug delivery, and medical device production [73]. Moreover, much of the biomass being used for collagen extraction is a by-product of fish processing for food, such as fish skin, scales, etc., thus offering both economic and environmental benefits [62]. Type I collagen is the most abundant type of collagen and can be found in fish scales, skin and bone, whilst type II collagen can be found in the fish cartilage [74]. The extraction methods need to be carefully studied and optimized to preserve the characteristics of collagen, because it is known that collagen extraction methods may directly influence its properties. Although the most commonly used extraction methods are based on the solubility of collagen in neutral saline solutions, acidic solutions, and acidic solutions with added enzymes such as pepsin [75], new extraction approaches are also being investigated, such as the use of ultrasound sonication, electrodialysis, and isoelectric precipitation, in order to maintain the intrinsic characteristics and biological properties of native collagen [38] (Figure 4). Mammalian and marine collagen differ in the imino acid content, in particular, the levels of proline and hydroxyproline vary significantly among fish species [76,77]. Furthermore, the composition of imino acids, especially hydroxyproline, is altered by environmental factors, such as the temperatures in which the fish live, and this affects the thermal stability of the collagen [77]. Collagen obtained from warm-water fish, such as bigeye tuna and tilapia, had greater thermal stability than collagen derived from cold-water fish, such as cod, whiting, and halibut [78,79]. Thermal stability differences are related to the content or concentration of hydroxyproline, which plays an important role in interchain hydrogen bond formation, contributing to the stabilization of the triple-helix structure of collagen [80]. Unfortunately, the higher rates of degradation, lower mechanical strength, low denaturation temperature and lower biomechanical stiffness, represent disadvantages associated with the use of marine collagen. The latter is, in fact, unable to form inter- and intra-chain bonds as mammalian collagen does, except through the use of chemical agents, and this represents a limit to the total replacement of mammalian collagen with marine collagen in biomedical applications. To overcome this obstacle and improve the physicochemical characteristics of marine collagen, it can be used as a blend with other biomaterials.

## 5. Use of Marine Collagen in Blends

A growing number of biomimetic composites based on collagen from marine origin are being developed and display very promising properties [81]. Marine biomaterials based on polysaccharides (chitin, fucoidan, alginate, etc.) and collagen are currently being thoroughly investigated for the biomedical industry, thanks to their biocompatibility in many applications and their specific properties. Collagen is generally widely used in various sectors; it is possible to improve its characteristics and mechanical properties, to make it more suitable for use in biomedical fields. Blends can be prepared based on synthetic and natural polymers, leading to the development of a new class of materials (biocomposites), characterized by improved mechanical properties and biocompatibility, when compared with materials of a single polymer [82]. In this regard, much research has been undertaken on chitosan, marine collagen- and alginate-based composites. Chitosan can be combined with collagen [83], or with collagen and hydroxyapatite [84,85]; chitosan/collagen cross-linked hybrid scaffolds possess better mechanical and degradation properties than their unitary systems separately, especially at the optimal ratio of 50/50. Thanks to their good cytocompatibility, those scaffolds can be used for the nerve tissue regeneration, as they promote the attachment, migration and proliferation of Schwann cells [86]. On the other hand, the combination of jellyfish collagen with functionalized denatured fish collagen (gelatin) permits the generation of collagen-based marine hydrogel (MCh), which has higher stability and is useful as an injectable cell delivery system for the cellular repair of cartilage in clinical applications. 

Shark-skin collagen is known to promote osteoblast growth and collagen synthesis in bone cells [87]. When this collagen is further mixed with the calcium phosphate of shark teeth to form a 3D composite scaffold, it better supports the attachment and proliferation of osteoblast-like cells [88].

These and other examples demonstrate how the use of blends can support the poor mechanical properties of marine collagen, improving its important biological functions (Figure 5).

## 6. Marine Collagen and Pharmacological Applications

In addition to the numerous biological properties of marine collagen, such as the ability to stimulate the regeneration of tissues such as bones and skin, and to give elasticity to tendons, etc., collagen hydrolysate and its low molecular weight peptides also show remarkable therapeutic properties and are of great interest for pharmacological and biomedical applications, as an excellent alternative to mammalian collagen. Many researchers have found that low-molecular-weight collagen peptides (di- and tripeptides), particularly those with Pro or Hyp C-terminal residues, have numerous bioactivities, including antibacterial [89], antioxidant [90], anti-inflammatory, immune-modulatory [91] and ACE-inhibitory properties [92,93]. Furthermore, another important property of collagen peptides is the ability to regulate lipid metabolism and cholesterol levels, with lipid-lowering and antiobesity effects [94], and to improve wound healing, applying tilapia collagen to a wound stimulated healing by increasing keratinocyte proliferation, fibroblast and myofibroblast differentiation, and ECM production [95]. Morishige and coworkers have demonstrated that collagen extracts obtained from the giant edible jellyfish *Nemopilema nomurai*, stimulate the production of immunoglobulins (Igs) and cytokines by human hybridoma cells and peripheral blood lymphocytes. Their data showed an in vivo immunostimulatory effect of jellyfish collagen, without any allergic complications [96]. Preliminary data from an in vitro study, demonstrated that the purified peptide fraction of collagen, obtained from fish skin hydrolysate, had an interesting anti-inflammatory property in the cellular microenvironment and could be used as a nutraceutical supplement [97,98]. Tomosugi and coworkers have shown that, in a rabbit model, the oral administration of the tripeptide component of collagen (CTP) results in its selective absorption into connective tissues; according to the results of these reports, CTP should have utility as a functional food for the prevention and treatment of atherosclerosis [99].

Decreases in atherosclerotic plaque area, serum total cholesterol levels, and the numbers of macrophages and smooth muscle cells in atherosclerotic plaques, have also been observed [99,100]. Furthermore, Gly-Pro-Hyp, the main component of CTP, inhibits dipeptidyl peptidase-IV activity, indicating that CTP may have potential utility in the prevention of diabetes [101,102].

There are several supplements in capsules used for the oral administration of fish collagen peptides, mainly used to counteract aging processes, in the cosmetic field. Low-molecular-weight collagen peptides can cross the intestinal barrier and enter the bloodstream in significant quantities. They can be absorbed intact from the intestine and resist degradation by plasma peptidases. This ability gives collagen peptides an effective biological function since they reach the target sites in an active form [103]. Following oral intake, collagen peptides are known to increase fibroblast production, activate multiple biochemical pathways, including the increase of hyaluronic acid production in dermal fibroblasts, and improve the water content of the skin. Commercially available marine collagen peptides have recently been clinically proven to be safe and effective for skin beauty, particularly when combined with skin-targeted plant-derived antioxidants, such as coenzyme Q10 [59]. However, oral administration of these peptides also shows other beneficial effects on the body. Marine collagen has been reported to reduce the production of proinflammatory cytokines such as COX-2, NO, MMP-13, and CTX-II [104], and is able to protect thymic epithelial cells (TECs) from cytotoxic and oxidative damage induced by cisplatin administration, which is used in the treatments of many cancers, due to its inhibition of the MAPK signal transduction pathway [105]. In addition to their antioxidant, antithrombotic, anticoagulant, anti-inflammatory, antiproliferative, antihypertensive, antidiabetic, and cardioprotective properties, marine-sourced biocompounds have been investigated for their neuroprotective potential [106]. Neurodegeneration is a complex, progressive multifactorial process that leads to the loss and death of neuronal structures in the nervous system; it is associated with the accumulation of insoluble deposits of protein and peptide aggregates, generally containing misfolded proteins, in different areas of the brain and spinal cord [107]. As oxidative stress has been considered to play an essential role in the onset and progression of neurodegeneration, there has been a significant scientific focus on the development of antioxidant compounds for neuroprotection [108]. Marine polysaccharides, in particular, chitosan derived from the hydrolysis of chitin, alginate extracted from algae [109,110], marine glycosaminoglycans such as hyaluronic acid (HA), chondroitin sulfate (CS), heparin and heparan sulfate (HS) obtained from mussels, codfish bones, tuna eyeballs, and shark fins [111,112], and marine glycoproteins [113] and glycolipids [114,115], are the most physiologically important biocompounds involved in neuroprotective activities that can be extracted from marine sources.

It is therefore evident that marine collagen peptides exhibit interesting biological activities and can be efficiently used for the treatment of various pathologies: skin ageing, bone defects, sarcopenia, wound healing, periodontal disease, gastroesophageal reflux, osteoarthritis and rheumatoid arthritis [116].

The abundance of collagen in marine organisms (fish, starfish, sponges, jellyfish, etc.), the resistance of some collagen-derived peptides to gastrointestinal digestion, as well as their ability to reach the bloodstream intact [59], suggest that marine collagen could be an interesting source of bioactive peptides, with promising pharmacological applications as functional foods and in drug delivery.

## 7. Marine Collagen and Biomedical Applications

The polymeric nature of collagen, as well as its low immunogenicity, makes it an excellent material for use as a scaffold, and for the production of smart nanocarriers for gene and/or cell delivery for tissue engineering applications. The biocompatibility between human and jelly-derived collagen suggests applications in the stimulation of tissue regeneration. The polymers used for scaffold preparation are mainly of biological origins, such as chitosan, silk fibroin, hyaluronic acid, collagen, and many others [117]; however, as mentioned above, the combination of organic and synthetic polymers gives rise to scaffolding characteristics combining the mechanical strength of synthetic materials and the biocompatibility of the natural materials. The most common synthetic biodegradable polymers in medical applications are poly (α-hydroxy acids), including poly-glycolic acid (PGA), poly-lactic acid (PLA), and polydioxanone (PDS). PLA, PGA, and their copolymers have been investigated for a higher number of applications compared with other degradable polymers [118,119,120]. The high interest in these materials is based on their good processability and mechanical properties, but above all on their ability to safely degrade without releasing toxic compounds, and on their high degree of biocompatibility [121]. Many synthetic polymers are used to produce three-dimensional and highly porous scaffolds, used specifically for bone and cartilage regeneration [122,123,124,125]. Bone tissue engineering is a promising strategy for the treatment of bone-related disorders, including osteoporosis and bone defects, promoting bone regeneration through the coordinated integration of stem cells, biomaterials, and bone-inducing factors. Although biomaterials represent the basic component of scaffolds in bone tissue engineering, they are often structurally modified or surface-functionalized to increase their potential. The functionalization of polymeric scaffolds with organic polymers, such as collagen or proteoglycans, is a promising approach to improve cytocompatibility. Organic polymers, isolated directly from the extracellular matrix, contain a multitude of surface ligands (fibronectin, laminin, vitronectin, etc.) and arginine–glycine–aspartic acid-containing peptides that promote cell adhesion. In tissue engineering, the combination of organic bioactive molecules and synthetic polymers gives rise to scaffolds characterized simultaneously by the mechanical strength of synthetic materials and the biocompatibility of natural materials [126].

Marine collagen also has applications in vascular tissue engineering. There is extensive evidence regarding the role played by the extracellular matrix (ECM), of which collagen Type I is among the main components, in driving capillary morphogenesis through sustained signaling, resulting in persistent endothelial cells (ECs) cytoskeletal reorganization and changes in cell form. A direct interaction between ECs and the ECM is necessary during angiogenesis, especially during the sprouting of new blood vessels from the existing vasculature [127,128]. Marine collagen, as well as mammalian ones, can contribute to the architecture and strength of tissues, interacting with cells through numerous receptors, and promoting cell growth, differentiation, and migration [123]. In a tumor context, collagen remodeling (degradation and redeposition) strongly influences tumor infiltration, angiogenesis, and migration; its reorganization at the tumor-stromal interface facilitates local invasion [129,130,131]. Paradiso and coworkers demonstrated that *R. pulmo* collagen is effective in manufacturing 3D devices, such as sponges, where it mimics the complexity of tissue architecture. OvCa cells migrated and differentiated within *R. Pulmo* collagen 3D scaffolds, confirming its suitability for advanced cell culture applications, and providing an excellent alternative to mammalian collagen sources for human cell culture [132].

Another biomedical application, in which the influence of collagen is widely documented, is the regeneration of cartilage tissues [133,134,135,136]. Cartilage is a connective tissue composed of sparsely distributed chondrocytes, embedded within a dense extracellular matrix primarily composed of type II collagen and proteoglycans. Unlike other tissues, articular cartilage is avascular and exhibits poor capability for self-repair; consequently, cartilage injuries are difficult to treat [137]. The transplantation of mesenchymal stem cells is a favorable approach due to their high proliferative activity and their capacity to differentiate into chondrocytes, which are responsible for cartilage synthesis and maintenance [138]. However, the most well-known cell-based repair strategy for large cartilage injuries is autologous chondrocyte implantation, which uses in vitro enriched chondrocytes from cartilage biopsy. Unfortunately, in monolayer culture, isolated chondrocytes lose their differentiated phenotype and shift towards a fibroblast-like phenotype [139]. Preserving the differentiated state, which ensures the ability to regenerate damaged cartilage, represents the main challenge during in vitro culturing. Three-dimensional (3D) scaffolds have the potential to preserve the phenotype of chondrocytes; particularly, hydrogels containing highly hydrated 3D networks are highly recommended because of their similarity to native cartilage [140]. Among these, the use of a collagen-based formulation of a marine hydrogel (MCh) has been proposed; it can cross-link with the cells trapped inside it, by in situ injection in the presence of H_2_O_2_ and horseradish peroxidase (HPR), without any cytotoxic effects. MCh can maintain the chondrocyte phenotype in vitro, compared to other 3D collagen hydrogels [66]. Although the data are still preliminary, MCh appears to have enormous potential as an injectable chondrocyte delivery system for cartilage repair, in both preclinical and clinical trials [66]. Many data in the literature indicate the use of marine collagen for a variety of applications, such as dental tissue engineering [141,142], oral mucosa regeneration [143], spinal cord injury repair and nerve regeneration [144,145], skin tissue engineering and wound healing [146,147,148], corneal tissue engineering [149], and many others. All these studies show that marine-derived collagen is a promising tool for wide-ranging clinical applications, including as a drug delivery system for cancer and other diseases.

## 8. Use of Marine Collagen for Drug Delivery 

Classical drug administration procedures (i.e., oral, intravenous etc.) present some limits due to the non-specificity of action and, therefore, the large amounts of drugs required to obtain a significant effect. The use of smart systems as vectors to specifically deliver drugs or other biological molecules (Drug Delivery Systems, DDS) can solve these problems, by increasing pharmacological efficiency. There are many different types of natural and synthetic micro- and nano-systems that can be functionalized for biomedical applications, including tissue engineering, antimicrobial techniques, bioimaging, theragnostic treatments, and vaccines [150,151]. In this context, it is possible to conjugate targeting molecules (such as folic acid for tumor therapy) [152], fluorescent probes for bioimaging analysis, or specific stimuli-responsive linkers (pH or redox-state induced) to release cargo in a controlled manner [153,154,155,156]. Many natural polymers such as inulin [157] can be used to produce controlled drug-release microparticles, replacing synthetic polymers, such as PVP, currently used for this purpose [158,159]. Collagen-based biomaterials have sparked great interest in the scientific community due to their unique properties, including high biocompatibility for several applications, low antigenicity, long-term stability both in in vitro and in vivo systems, and the possibility of functionalizing them with -OH, -NH_2_, -COOH and -CH_3_ groups [128]. Furthermore, their nature permits modulation of their shape and size, depending on the specific application [160]. It is possible to synthesize collagen microparticles (MPs), sponges, hydrogels or scaffolds, and to combine them with molecules of different kinds, in order to create a controlled drug-release mechanism [42] (Figure 6). Marine collagen microparticles can be easily synthesized (e.g., by an emulsification-gelation-solvent extraction method) and the matrix organization permits the entrapment of cargo and subsequent cross-linking. In this manner, cargo release can be obtained by erosion of the collagen compound, inducing a two-stage release: an initial burst release (of the biological molecules loaded in the surface area), followed by a slower release rate, lasting a few days [161]. The loading capacity and release rate can be increased by incorporating microparticles of collagen/gelatin/hydroxyethyl cellulose, etc., into polymeric scaffolds [162]. It is also possible to incorporate MPs of other kinds (e.g., polymers) into fish collagen-based scaffolds, in which the higher swelling and degradation rate enhances the diffusion rate of the MPs, and consequently of the proteins. From this point of view, MPs can be loaded with different types of proteins, including growth factors such as bFGF, whose release depends on the size of the MPs and the amount of collagen in the scaffold [163]. 

At the same time, collagen hydrogels or sponges are amply adopted as delivery systems, thanks to the possibility of regulating the network and mesh size, and therefore controlling the drug release: the smaller molecules can diffuse through the hydrogel, while the larger ones can be trapped, and released after the erosion of the matrix [164,165]. 

For this reason, collagen-based materials are widely used in the treatment of surgical wounds to prevent bacterial infection. Although mammalian collagens (bovine, equine, and porcine) are the most used collagen types for tissue engineering applications, as mentioned above, they present some limits due to the possibility of inducing allergic reactions, the risk of bovine spongiform encephalopathies, and ethical and religious limitations [166,167]. On the other hand, marine collagen, for example, tilapia collagen, is less antigenic than mammalian, and can exist for a long time in an organism without producing a severe immune response [168]; marine collagen is unable to induce the transmission of ruminant zoonoses, and can promote cell adhesion and differentiation [169]. Furthermore, the disadvantage of the lower denaturation temperature can be overcome by adding cross-linking agents such as carbodiimide, as proposed by Lukáč and colleagues [170]. They produced cross-linked collagen sponges from freshwater fish skin (*Cyprinus carpio*) to deliver gentamicin for the treatment of surgical wound infection in a rat model. Depending on the bacterial infection, the same marine collagen functionalization can be adopted for the treatment of biofilm-related infections of bones [171], or for the delivery of other antibiotics such as vancomycin for wound dressing [172]. 

In this regard, Langasco et al. have developed a natural marine-sponge collagenic skeleton loaded with L-cysteine hydrochloride as a DDS in chronic wound treatment. The collagen matrix permits the adsorption of the excess wound exudate, and gradually releases the drug which stimulates the re-epithelization process [25]. Therefore, because of its role in cellular growth and proliferation, marine collagen is widely used for wound healing applications, sometimes in combination with hydroxypropyl methylcellulose, or with natural molecules involved in the prevention of oxidative damage, such as curcumin [173]. In vivo studies have demonstrated that the combination of curcumin with collagen can increase wound contraction activity, without causing irritation, suggesting its possible use in dermatological applications [174].

The combination of marine collagen with compounds such as carbohydrates, can increase its properties as a drug delivery system, as proposed by Nguyen and colleagues, who combined collagen derived from Vietnamese freshwater carp fish scales with carrageenan, to obtain pH-sensitive hydrogel beads. In this manner, they were able to improve the controlled bioavailability of loaded allopurinol for gout treatment [175]. 

On the other hand, fish collagen can be combined with chitosan to form a film for the delivery of doxorubicin-loaded nanoparticles (NPs), as shown by Chen and colleagues. In this manner, they obtained a double-controlled drug release mechanism. The collagen film degradation and swelling led to greater NP release, followed by increased drug release [25]. The doxorubicin release can be further regulated by modulating the ratio between collagen and chitosan and is related to the pH of the release medium [176].

## 9. Conclusions 

Currently, marine collagen is gaining increasing attention due to its safety, biocompatibility in many biomedical applications, biodegradability, greater ability to penetrate lipid-free interfaces, and numerous biological activities and therapeutic properties, such as antioxidant and anti-inflammatory activity, neuroprotective, anti-ageing, and healing effects. Due to its enormous potential, it can be used as a sustainable platform for the biological valorization of fish by-products and as an alternative to animal collagens, for inclusion in cosmetic and pharmacological formulations. Collagen hydrolysate and its low molecular weight peptides can certainly be used for effective treatments against skin ageing, resulting in additional effects on the psychological and social well-being of individuals. Unfortunately, the higher rate of degradation and lower denaturation temperatures, represent disadvantages associated with the use of marine collagen, which today still limit the complete replacement of mammalian collagen in biomedical applications. Future research objectives could be focused on improving the physicochemical characteristics of marine bioresources, as increasing the denaturation temperature of different types of marine collagen could be fundamental to broadening its effective applications in the biomedical field, and on their potential in drug delivery systems, for the specific release of drugs or other biological molecules, increasing the effectiveness of drugs.

## Figures and Tables

**Figure 1 molecules-28-01152-f001:**
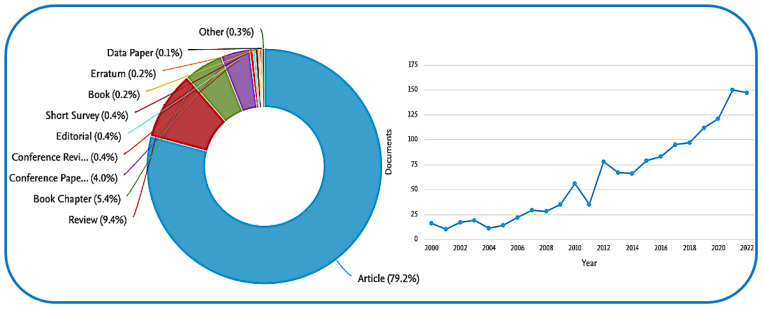
The trend in the number of publications related to marine collagen indexed on Scopus, from 2000 to 2022 (last accessed 29 December 2022 https://www.scopus.com/term/analyzer.uri?sid=8346601bf23e38fdb98df3827ad94193&origin=resultslist&src=s&s=ALL%28marine+collagene%29+AND+PUBYEAR+%3e+1999&sort=plf-f&sdt=b&sot=b&sl=40&count=262&analyzeResults=Analyze+results&txGid=d83dade43155068dbf08353305d81e52).

**Figure 2 molecules-28-01152-f002:**
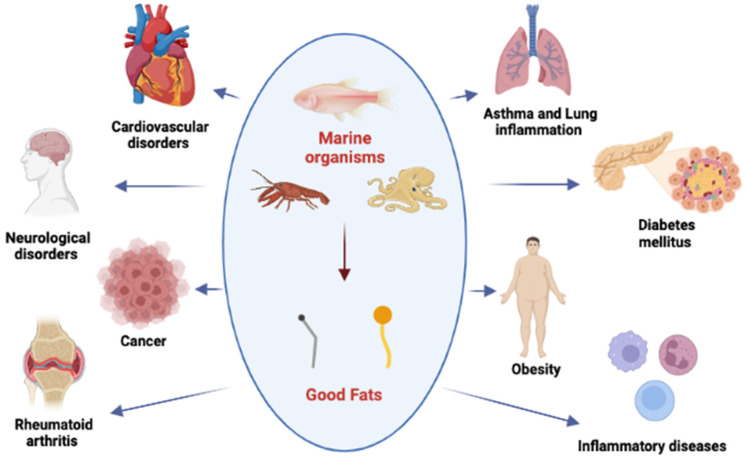
The intake of “good fats” has many beneficial effects in the prevention and treatment of different chronic diseases. Created with BioRender.com.

**Figure 3 molecules-28-01152-f003:**
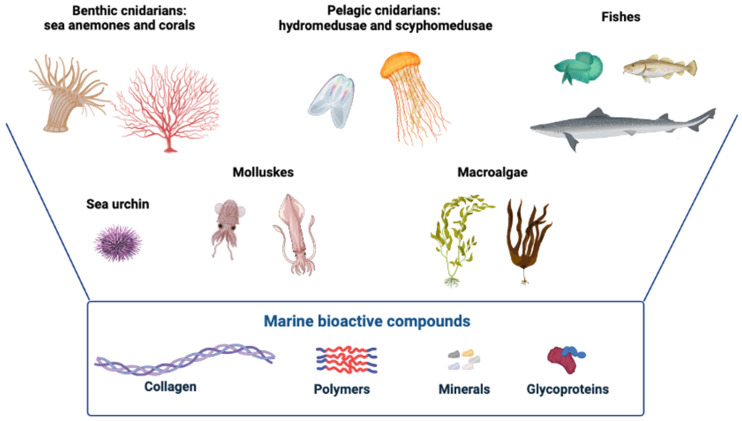
Main sources of marine bioactive compounds (collagen, polymers, minerals, and glycoproteins): cnidarians, fish, molluskes and, microalgae. Created with BioRender.com.

**Figure 4 molecules-28-01152-f004:**
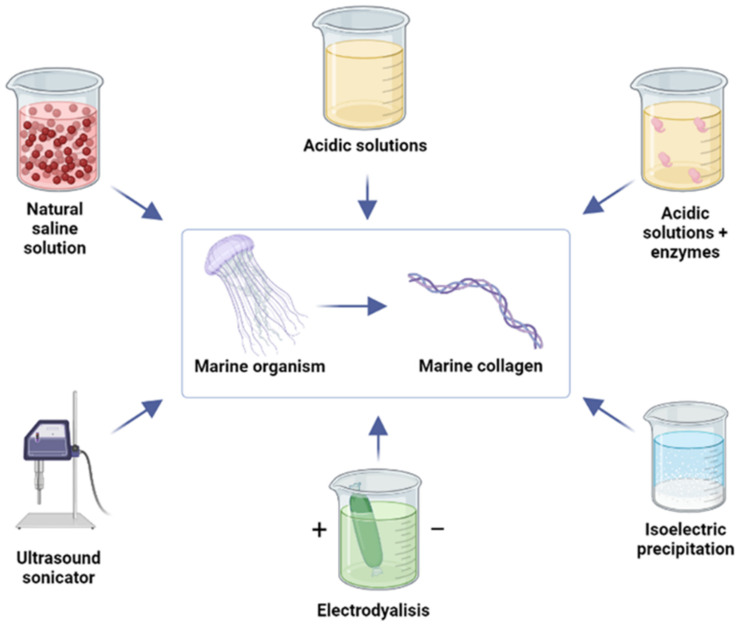
Collagen extraction methods: neutral saline solutions, acidic solutions, acidic solutions with added enzymes, ultrasound sonication, electrodialysis, and isoelectric precipitation. Created with BioRender.com.

**Figure 5 molecules-28-01152-f005:**
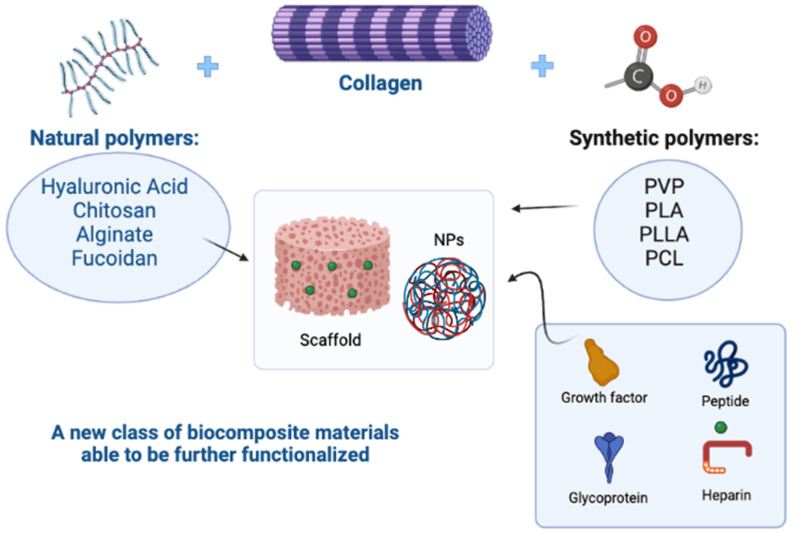
The use of synthetic and natural polymers in blends has led to the development of a new class of materials that support the poor mechanical properties of natural polymers, improving their important biological functions. In addition, scaffolds and nanoparticles (NPs) can be further functionalized with glycosaminoglycans such as heparin, glycoproteins, peptides, or growth factors, to improve the biocompatibility of the devices (Polyvinylpyrrolidone-PVP; Polylactic acid-PLA/PLLA; Polycaprolactone-PCL). Created with BioRender.com.

**Figure 6 molecules-28-01152-f006:**
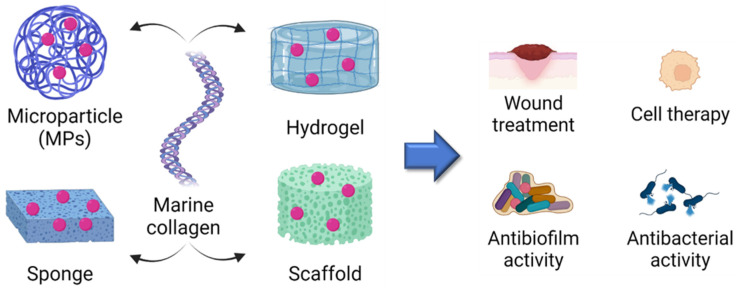
Marine collagen microparticles (MPs), sponges, hydrogels and scaffolds, and their application as delivery systems (wound treatment; cell therapy; antibiofilm and antibacterial activity). Created with BioRender.com.

## Data Availability

Not applicable.

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
