# Peer review of "Recovery of Bioactive Compounds from Marine Organisms: Focus on the Future Perspectives for Pharmacological, Biomedical and Regenerative Medicine Applications of Marine Collagen"

_molecules, 2023, doi:10.3390/molecules28031152_

Round 1

Reviewer 1 Report

I can recommend that this Review is suitable for publication, although first it does require some significant attention prior to publication. 

There are 5 recurring items that need attention.   Three are typographical and readily corrected in the many places that they occur.

1.   Capitalization.   There are some words in the text that have a capital letter where a lower case is more appropriate.   Some, but not all, are noted later.

2.   Italics.   On several occasions when species names are given, they are not in italics and the second part of the binomial is capitalized rather than lower case.

3.   Abbreviations.  In some places an abbreviation is defined, but it is never used subsequently, so it should be deleted.

 The other two need significant attention.

4.  Biocompatible.  This descriptor is regularly used as a generic property of a material.  This is not correct.  Biocompatibility is a property when used in a specific application, and not a general property of a material. (See the Williams dictionary of Biomaterials and subsequent definitions.).   For example, while collagen may be biocompatible for wound repair, it is not biocompatible for blood contact where thrombosis is triggered, nor in load bearing situations.  Similarly, while a metal or ceramic may be ideal for hard tissue repair, they are hardly suitable or biocompatible for wound repair.  So, biocompatibility should not be used as a generic term.   The text needs rewording such that the collagen could be described as ‘biocompatible in a number of contexts’, or something similar, rather than as an overarching, somewhat meaningless statement. 

5.  The References.   It looks as though a computer program has been used and that trust in the program has been misplaced.  And the reference list has not been proofread before submission.  There are various groups of problems.  (a)  Abbreviated or not abbreviated journal names.  For example, Mar Drugs or Marine Drugs, and many others.  Check the Journal style and be consistent.  (b)  No volume detail or page numbers.  In many cases, the citation does not include any page number information, and in some cases the volume information is also missing.  For example, references 1, 26, 33, 37, 41, 44, 68, and so forth.  Many.  (c)  Duplication of details. In some cases, the volume and page details are included in the italics journals title and need deleting.  For example, 21, 23, 42, 59, 66, and so forth.   (d)  Italics and capitalization of species names.  These need validating against the original articles, not an abstract.
These are repetitive issues.   Individual issues are discussed later.

 There are several other points in the manuscript.

L 61 to 65.  Is this big gap intended?

L 67.   Figure 1.  A good figure but needs correcting.  The apparent decrease recently is an aberration?  A data point of zero for 2023 should be deleted.  How to handle an incomplete year for 2022?   Possibly delete, possibly update before final submission, or possibly extrapolate to the full year based on the date when the numbers were downloaded.  Some correction is needed, and the legend adjusted if necessary.

L 106   Figure 2.   “Immune system” seems out of place relative to the other items mentioned.  Would ‘inflammatory diseases’ or something else be better?

L 118.   Jellyfish becomes a major topic.   How reliable is it as a commercial material.  Certainly the “blooms” that now occur regularly can provide a source but is it regular and reliable throughout a year.  The frequency of blooms has certainly increased over the last few decades, potentially due to ocean temperature change and/or environmental runoff.  I think it unlikely that this will be brought under control in the near future.  Nevertheless, some extra comment on supply issues could be useful.  And the conundrum that improvements are a potential negative for commercial exploitation.

L 150.   Are cephalopod fisheries a useful source, equal or better than molluscks for example.  Possibly include?

L 157.   “… vertebrates, where it often fulfills an important mechanical role, while also being involved in a range of key cellular and molecular interactions [32 ….”     That is collagen should be seen as the full family of proteins in this context.

L 162.   There are 28 types.  Check your reference 36.   The original claim for a type 29 related to a fragment that later proved the same as a previously described collagen.

L 164.   This sentence may need expanding for clarity.  See above regarding ‘biocompatible’.  Gelation capacity may need defining better so as to describe the property of extracted purified collagen as opposed to the denatured form, gelatin.

L 177.   Is there evidence of FMD transmission via collagen, or is it speculation.  Check the reference else add another that confirms this.

L 184.    ‘not’    This needs correcting as products based on recombinant collagen are marketed.  See, for example,   https://collplant.com/products/commercial-products/      There may be others too.  “not yet adopted”   could be  ‘has only been infrequently adopted’  or similar.

L 193. Italics for species.

L 196.   Biocompatibility is not used correctly.    Biocompatible is selected clinical applications…

L 199.     used as a biomaterial for prostheses,   ?

L 204.    .. fish cartilage. 

L213.    Proline and hydroxyproline are  imino  acids.  This can be used where the text is specifically about these, rather than the more general texts of overall compositions.

L 215.   Should ad be as ?    However, this suggestion is not recent as implied by your reference 71 and citing the author by name.   Rather it goes back a long way, to the 1970’s by scientists such as BJ Rigby and M Mathews, who reviews it in his 1976 book.  Perhaps give credit to the pioneers?   So, both imino content variation as well as the degree of hydroxylation are both important.

L 221.  Although mentioned later, is it worth noting here that the lower thermal stability is not an issue when products are crosslinked to the extent that takes their stability above human body temperature.

L 262.   I was puzzled by my first look at this figure.  I wondered how the 4 items at the bottom related to the top half.   Perhaps an arrowhead from the bracket up to the Biocomposite Materials text would clarify this?

L 277   that low     no comma?

L 278.     Delete    shown    to make sense?

L 284.   Citing names needs to be consistent.  Here you use a given name, but elsewhere only surnames are used, eg L 291.

L 290 and onwards.   It is unclear when the findings related to properly controlled clinical trials or not.  Emphasis should be on such, and this stated when it is the case.  If not, the evidence is an anecdote?   Many make anecdotes, not data.   If the evidence is not from a full trial perhaps add a comment such as ‘preliminary evidence suggests’. 

L 327   especially chitosan, derived by hydrolysis of chitin, and alginates ….

L 328    hyaluronic

L 334-6    Too many capital letters

L 380.   References to support these claims are needed in the specific places.

L 384    Italics species

L 411    A reference (or two) is needed to support this last sentence.

L 420   Collagen    No e

L 435    Again, careless use of ‘biocompatibility’

L 436-7    Numbers as subscripts

L 439.     See Williams dictionary of biomaterials.   Scaffolds are external structures, while templates are 3-dimensional?

L 462.   “marine collagen is less antigenic”    I was not aware of any detailed immunological studies on marine collagen, but may have missed it.  It is important to include the reference{s} on this key aspect for biomedical applications.  Indeed, if the marine collagens are used without adequate stabilization, the gelatin that forms on denaturation could prove to be distinctly immunogenic as new epitopes are now being presented.

L 476     HPMC    Not used elsewhere?     There are others like this.

L 500    Again, incorrect use of ‘biocompatibility’

L 509    I am sure that not all Jellyfish will give a collagen that has a melting temperature of 29C.  It will depend, as you noted before, on the environmental temperature.  Thus, estuarine species, often used for food, will probably differ from oceanic species.  A range may be more appropriate, of the details of the specific species and the Reference included.

L 526.  As noted at the start the Reference need significant attention.  Many issues occur on multiple occasions.  Nevertheless, the overall extent of citations is very good, with only a few examples as noted above needing addition.   Careful proof reading is needed else it will reflect badly on the authors.  Some additional comments (not necessarily complete) on individual items follow.

1.  Page number(s) missing.    This occurs frequently.

2.  Journal title missing.  Or if a web citation the access date missing.

3.  An example of a Journal title not abbreviated

41.   should it be Class I and II ?

46.  This reference citation brings up an article by MT Beregeon in the  J Okla State Med Assoc from 1967.  Something, a Book or a review, that is more current and readily available, 2018 onwards preferably, may be more appropriate.

50.   C, E.: and G, M.  names in full would help.

54.  An example where the is no Journal abbreviation consistency (elsewhere Mar Drugs) and additional text follows the name prior to the date and the other data.

67.   The reference is duplicate of #54.   Is this the only case of duplication?

85.    Was species in italics and with lower case e in the original paper?

86.    Ditto.

119.    The Journal title needs adding;  currently given as undefined !

146-149.  More examples of missing page numbers!

159.     Seems incomplete.  If an article needs details, if a book needs publisher.

Finally, as well as the major attention noted for the References, could the authors please check that using capital letters in the titles of Journal articles is correct for the style normally used by the present Journal.  (Personally, I find it more difficult to read.)

Reviewer 2 Report

I have some recommendations to improve the paper:

Figure 1: The font size is too small.

Page 4, line 123: It would be beneficial for the reader to have some examples of natural cross-linkers.

Chapter 4: Regarding collagen origin. What about poultry collagen? Is it worth mentioning?

Figure 5: Glicoprotein – use “y” after “l” – Glycoprotein.
